# Dante and Siger: An Intellectual Mission Overcoming Error and Authority

Annalisa Guzzardi 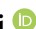

The Graduate Center, City University of New York, New York, NY 10016, USA; aguzzardi@gradcenter.cuny.edu

**Abstract:** This article places the question of the presence of the Averroist philosopher Siger of Brabant in Dante's *Paradiso* within the debate concerning Dante's relationship with Averroistic philosophy throughout his works, particularly in the *Commedia.* The purpose of this research is to initially query the issue of Dante's potential retraction of previously adopted Averroistic positions in the *Commedia*, and then to examine in greater detail the relationship between the Florentine poet and the Brabantian philosopher. United by the thematic and autobiographical thread of knowledge persecuted by power, embodied in the history of both figures, the proposed viewpoint is that Dante finds a model of intellectual honesty and freedom in Siger's thought, ready to question even the authority of Aristotle, a model that Dante admires, places in Paradise, and upon which he structures his *poema sacro*. In conclusion, this work suggests a reading of some paradigmatic and antithetical pairs in the *Commedia*, linked by common principles yet opposite destinies, including Dante himself.

**Keywords:** Dante; Siger of Brabant; Averroism; philosophy; knowledge; power; synthesis; authority; Aristotle; truth





## 1. Introduction

The proposed study moves within the complexity of the presence of the Averroist philosopher Siger of Brabant in Dante's Paradise to elucidate a portion of Dante's relationship not only with Averroistic thought throughout his poetic production but more specifically the role that this relationship plays in the *Commedia* and Siger's placement in paradise. I begin by outlining the relationship between Dante and Averroism and the contact that he may have had with the dispute about the qualities of the intellect between the Faculty of Arts and the Faculty of Theology. Then, I explore the relationship between philosophy and happiness established by Averroistic thought in greater detail.

Between the *Convivio* and the *Commedia*, I explore how Dante used Averroism to reflect on the knot of the highest rational knowledge attainable through human faculties, always trying to contain the impulse of a desire that transcends human possibilities and closes in on itself presumptuously. On the one hand, this involves not indulging in a strictly 'progressive' or 'evolutionary' linear interpretation of the journey from the *Convivio* to the *Commedia*. On the other hand, it is important to acknowledge the moment of strong discontinuity implied in the perspective of the poem, driven by a desire much different from that which characterised purely philosophical epistemology in the treatise.

By focusing on the *Commedia* and the presence of the Averroistic motif between Limbo, the circle of heretics, Purgatory, and finally the Heaven of the Sun, I subsequently connect this autobiographical theme to the recurring motif of *ingegno*, which transforms from one canticle to another. By shifting to the relationship between Dante and Siger of Brabant and the difficulty of determining precisely through which textual paths Dante came to know the evolution of his thought and personal restitution of Averroism, I examine the symbolic role of the philosopher's violent death, which Dante was likely aware of, and how it places Siger within a lineage of persecuted philosophers and victims of injustice in which Dante positioned himself as the last representative.

However, if Dante identified in this way with an Averroist philosopher, it is necessary to examine the basis on which an intellectual consonance was established between the two—one that goes beyond the threat of knowledge by the ecclesiastical power. In this paper, it is proposed that this consonance can be identified in the distinction that the Brabantian makes between 'probable' and 'necessary' with regard to Aristotle's most radical statements, especially in asserting the fallibility of even the Aristotelian authority when the supernatural comes into play. This may be one of the elements of Siger's reflection that mattered most to Dante, especially in view of the construction of his new authority through the poem, an admission of the fallibility of the highest philosophical authority that simultaneously translates into a profession of humility on the part of the philosopher towards a higher and unfathomable knowledge.

To conclude this discussion, I propose in a brief 'ouverture' that, when evaluating a particular system of thought or area of knowledge, Dante assigns opposite destinies to individuals representing these ideas through antithetical pairs of characters who, on Earth, shared the same system of thought but, due to their differing internalizations and reaction to these systems, experience very different destinies in the afterlife. These presumed pairs textually interact in the *Commedia*, which creates complex intersections and chiasmi, some of which are perhaps formed by the combination of such peculiar destinies as those of Cavalcanti, Ulysses, Siger, and Dante himself.

## 2. Dante and the Averroistic Tradition

To examine in depth the presence of Siger of Brabant, an Averroist philosopher, in Dante's *Paradiso* and the poet's conception of his work in relation to philosophical and theological truth, it is fundamental to explore Dante's relationship with Averroism. This topic has occupied scholars for years, and much has already been said about it. To trace the literature on this matter, it is necessary to begin by tracing the influence of Ibn Rušd's (Averroes) commentary on Aristotle's works. Medieval thinkers adopted Aristotle's ethical, physical, and metaphysical systems to reconcile them with theology and integrate this pagan knowledge into the Christian world. Among the various interpretations of Aristotle's works, some lent themselves well to this recovery of his knowledge because they ironed out all potentially heterodox doctrines in such a way that there were no contradictions with the truths of faith.

However, the interpretation of the Arab philosopher Averroes posed a problem within this framework due to his theory of monopsychism[1], which denied the immortality of the individual soul and the personal nature of knowledge by asserting the universal sharing of the possible intellect and the eternity of the world and by denying that the intellectual soul was the substantial form of the individual. Therefore, cognitive activity could not be attributed to the individual. Embedded in a materialistic and rationalistic view of reality, the Averroistic tradition thus promoted a relationship with philosophy that allowed man to attain complete speculative happiness in earthly life via *continuationis*—that is, by gradually ascending through the stages of the process that can lead man from sensory experience to union with the knowledge of separate substances.

This conception of the soul, the world, and philosophical science strongly animated debates in French and Italian universities between the second half of the 13th century and the beginning of the 14th century. These discussions primarily involved the Faculties of Arts and Theology. However, as noted by Barański and Bianchi (Cf. Barański 2013; Bianchi 2014b), this cultural ferment cannot be simplified into a simple polarity of philosophers versus theologians or orthodox versus heterodox, as there was neither homogeneity in the definition of 'orthodoxy' and 'heterodoxy', nor did these fields clearly distinguish between purely philosophical or purely religious matters. Indeed, there was mutual contamination of knowledge between these areas.

Furthermore, Averroism involved ethical aspects, as it entailed a confidence in the ability of human reason to provide individuals with complete immanent happiness. This extended to the philosopher's role and significance; it even went so far as to support

a 'genuine 'divinization' project of the philosopher', conceiving the ultimate happiness attainable by the philosopher in earthly life as a result of their 'conjunction [*coniunctio*, *continuatio, copulatio*]' with separate intellects and God, understood in Averroistic terms. In this way, these thinkers would have fundamentally transformed the 'intellectualistic eudaimonism of the *Nicomachean Ethics*'[2].

Although Barański suggested that Dante would not have been convinced by these ideas[3] and that there is no evidence of that, the poet was certainly in contact with these notions about the autonomy and self-sufficiency of philosophical reason. First, these ideas were shared by his close friend Guido Cavalcanti. Second, these theories had made their way from France to Bologna among not only the masters of the arts but also theologians.

Moreover, what undoubtedly captured Dante's attention was the condemnation of 219 theses, including Averroistic ones, by the Bishop of Paris, Étienne Tempier, in 1277, along with 13 already condemned in 1270. Tempier emphasised the danger that they posed to the Christian faith, as they were being taught and disseminated by the masters of the arts in the Parisian academy. This event likely left a significant impression on Dante since an ecclesiastical authority had condemned a particular expression of philosophical science and its autonomy from faith. Thus, what was being attacked was a certain culture, which had significant implications for Dante. Tempier's theses not only included those by Siger but also figures such as Thomas Aquinas. Therefore, Dante wanted to distance himself from this ecclesiastical power, often used to silence enemies through excommunication and charges of heresy, and the academic approach to knowledge based on a divisive method.

However, achieving a better understanding of Averroism's significance in Dante's time is quite complex, as Bianchi illustrated[4]. The application of this category to medieval thought is retrospective and therefore risks falling into anachronism if not used with caution, as the term was coined in 1852 by Ernest Renan. Subsequently, it was simplistically associated with a philosophical doctrine that was diametrically opposed to Christianity under the name of radical Aristotelianism (or Latin Averroism). Even the 'followers of Averroes' did not constitute a unified and compact group that shared a specific doctrine as a homogeneous movement. Instead, they primarily shared the exegesis of Aristotle's texts proposed by Averroes and the logical consequences that followed in terms of philosophy and worldview. As highlighted by Bianchi, it was more of an 'intellectual attitude' among philosophers with diverse interests and inclinations of thought. Therefore, it would be more accurate to discuss 'Averroistic tradition' rather than 'Averroism' as a doctrine[5]. Within this tradition, there were those who advocated for the unity of the intellect, including Siger; they formed a diverse group of thinkers with various interpretations of the same ideal of philosophical knowledge as the fullest realisation of human potential and capable of bringing the highest degree of happiness to humanity.

Although Dante encountered these ideas and likely dedicated part of his own reflections to them, he did not necessarily adopt them as his own or accept them, and they should not be openly identified in his works as evidence that he underwent a certain 'Averroistic' phase. Instead, it would be more useful to imagine that Dante, with his anti-academic attitude and reluctance to divisions and differentiations between various types of 'truths' (philosophical and revealed), did not position himself in this controversy between the masters of the arts and theology from within but rather in a more transversal, complex, and varied manner.

It is important to bear in mind that, although Dante did not accept the idea of monopsychism or the divinisation of the philosopher, he may have been interested in how the masters of the arts elaborated on the tension between knowledge and happiness, the problem of the human ability to penetrate the divine through rationality, and their way of resolving this issue. Instead of adopting the doctrine of *copulatio*, many of these masters argued that the ultimate goal of man—the contemplation of the first causes—was achievable because one could satisfy their thirst for knowledge even if they came to know God in an incomplete manner in this life. Therefore, genuine happiness is possible when a person grasps the

existence, causality, and some properties of divine realities without claiming to know their essence or have an intuitive vision of them.

### 3. The Averroistic Thread in Dante's Works

The idea of earthly happiness is present in the *Convivio*, the philosophical work in which Dante explores both the potential of the human intellect and its limitations. When Dante examines the question of how humans can know God and the immaterial, he argues that this is only possible by starting from their effects, then without ascending to knowledge of the cause[6].

In this regard, there is a famous passage from the *Convivio* (III, xv, 6–10) in which Dante arrives at the logical conclusion that, if nature does nothing in vain and humans do not have the cognitive ability to attain knowledge of separate substances, God, and eternity in their essence, then they do not even have a natural desire to know them; it is only in this way that man can attain full earthly bliss. This assumption, which is somewhat related to what was mentioned earlier about Averroist thinkers, has sometimes been understood as an involution and withdrawal of philosophical knowledge into itself when realising that it cannot aspire to a reality beyond human capabilities.

It seems more reasonable that Dante, while reflecting on the role of philosophy and the possibility that its knowledge can make man happy in this life, aligned himself with philosophers who did not desire to transcend their rational capabilities and directly attain intuitive knowledge of the divine and its cause, as mentioned earlier. He promoted a vision of philosophical knowledge that stays within its limits concerning what can be known in this life and does not direct its desire towards an unattainable end by these limited means. On the other hand, if this in some way expresses a kind of proximity to the positions of those philosophers who, within the Averroistic tradition, had moderated their views and recognised the limits of reason, it also represents a departure from the doctrine of Averroes understood as the possibility that man can, through intellect, be in conjunction with separate substances and attain the highest knowledge of divinity.

Fioravanti emphasised that, by presenting this 'participatory beatitude' of man through earthly knowledge, Dante did not associate its imperfection with a simple 'deficiency' but rather understood that 'a natural desire can only be satisfied in a natural manner'. Therefore, philosophy satisfies the natural human desire for knowledge, a desire in whose name Aristotle opened his *Metaphysics*: '*omnes homines natura scire desiderant*', which in the *Commedia* will meet the Augustinian '*fecisti nos ad te et inquietum est cor nostrum donec requiescat in te*' (*Confessions* 1.1). With Fioravanti, it is possible to affirm,

> The limitation sets a limit, it is true, but it also creates a full measure, without any leftovers; if nature has not given us the possibility of knowing the essence of God and separate substances, this means that there is no natural desire for this knowledge in us. (Fioravanti 2014, pp. 41–42)

Dante aimed to emphasise that every element of knowledge is perfect in itself; thus, the progression from one element to another is not a sign of imperfection but rather increasing perfection. In this way, in the *Convivio*, Dante limits himself to discussing philosophy and defending its autonomy and completeness while tempering its claims. However, he does not achieve this by adopting Averroistic anthropology (the philosopher who is capable of accessing the essence of divinity through *copulatio*). As Porro argued, Dante used an Averroistic argument but reversed it to defend the possibility that natural knowledge can bring happiness to humanity (Porro 2010, pp. 655–57). This allowed him to justify the openness of his work, which in its mission of vulgarisation encouraged trust in the legitimacy of scientific knowledge and the promise of happiness that comes from it on Earth. Moreover, this mission was primarily extra-academic, in contrast to institutionalised and elitist knowledge.

In the *Commedia*, desire undergoes a transformation and can transcend the limits of reason; indeed, it must do so. In this work, the objectives and perspectives change compared to the *Convivio*, and the positions presented there are surpassed. Desire itself assumes

a very different connotation, given that the poet presents its first manifestation as eternal damnation in Limbo. In this context, desire, when misdirected—by, for example, philosophers who believe that they can extinguish their desire within the realm of philosophical knowledge—becomes, by contrast, what souls must eternally endure without hope of extinguishing it, which causes them eternal suffering. However, as highlighted by Porro, there does not seem to be a 'radical distancing from the previous adherence to Averroism' between the *Commedia* and the *Convivio* on Dante's part. Indeed, Dante was never strictly an Averroist, even with regard to the question of the natural desire for knowledge.

In any case, the prevailing interpretation of the journey from the *Convivio* to the *Commedia* is that Dante was inclined to support Averroistic positions in the *Convivio* to a certain extent, at least with regard to the independence and self-sufficiency of philosophical knowledge in relation to matters of faith and humans' potential to attain full happiness by reaching the pinnacle of philosophical speculation. These were the ideas that had an Averroistic flavour in the treatise. Some scholars, such as Nardi, have perceived a three-stage evolutionary process from Dante's *Convivio* to his *Monarchia* and finally the *Commedia*, in which even the vaguest Averroistic assertions would fit within the realm of perfect Christian orthodoxy[7]. On the other hand, Freccero viewed the incompleteness of the *Convivio* as a sign of the failure of the non-conformist ideals that it promoted and described the work as a 'shipwreck' (Freccero 2014, pp. 51–52). He believed that the *Commedia* marked a true doctrinal and ideological leap, with a strong discontinuity: 'Dante began the journey of the Convivio hoping to reach a safe harbour. The opening of the Divine Comedy portrays the pilgrim as a shipwreck, washed ashore from the pelago on which he had set forth' (Freccero 2014, pp. 51–52).

Furthermore, Freccero viewed the character of Dante's Ulysses as a representation of the philosopher who is characterised (and eternally damned) by the uninterrupted exploration of earthly knowledge beyond its true limits. He also perceived this characteristic in philosophers such as Boethius and Siger, who were nevertheless eternally saved and praised by Dante[8]. Even in the *Convivio*, Dante wanted to test those limits (Barolini 2014, p. 266), explore that boundary as far as possible, but always trying to stay this side of the 'Averroistic hubris'. He consistently emphasised this boundary in a way that suggests that, even at that point, his concerns were akin to those of the *Commedia*. There was not a full adherence to Averroistic doctrine, but rather a clear differentiation that is shared with the *Commedia*, placing the poem in continuity with Dante's doctrinal thought in the *Convivio*, but staging a different ideological mission which involves the leap and fracture pointed out by Freccero.

It is worth noting that Barański questioned the interpretation of the *Convivio* and the *Commedia* as ideologically opposed, denied the evolutionary perspective, and reminded us that the *Commedia* was not perceived as completely orthodox. Indeed, medieval commentators had to defend Dante from accusations of heterodoxy and argued that his work was allegorical poetry and should not be taken literally. In Barański's words, allegoresis was a 'tool of extraordinary ideological flexibility' and a way to normalise ideologically discordant and disturbing elements[9].

In the *Commedia*, the poet indeed chose to position himself in a way that allowed him to surpass that limit while exorcising the dangers of such an enterprise. He did not do this by relying on rational philosophical knowledge that was typical of the masters of the arts but rather the supernatural illumination of divine grace. Indeed, Dante was clear in his rejection of the possible unity of the intellect, both in the *Convivio* (IV, xxi, 5) and in *Purgatorio* (XXV, 61–66), and condemned the heretics '*che l'anima col corpo morta fanno*' ['all those who say the soul dies with the body']ial[10].

As suggested by Barański, the *Convivio* could be seen as an exposition of how reason can be correctly directed not only towards the ethical and intellectual progress of humanity but also opening a window on the transcendent[11]. It is a work in which the author aims to bring knowledge to the entire community, not only elite philosophers. The *Convivio* was part of a long-term project that was ultimately fulfilled with the *Commedia* and aimed to

syncretically sew together fragments of truth to finally have a complete picture of it that transcends the incompleteness of earthly knowledge. It is interesting to note that, according to Bianchi, this cultural dissemination project opposed the teachings of Averroes, who deemed it dangerous to allow everyone to access knowledge. Similarly, Averroes would have found the endeavour of the *Commedia* absurd, with its mission of revealing eschatological truths to the people by mixing different levels of discourse such as philosophy, poetry, and theology, which, according to the commentator, should be kept distinct[12].

Similarly, even the *Monarchia*, which has been interpreted with elements of political Averroism regarding the theory of the two ends of humanity, in line with the alleged theory of the 'double truth'[13], presented a contrasting ideal to that of Averroistic influence. Indeed, in Dante's work, the theory of a universal imperial power that guarantees peace for humanity is founded on the premise that the entire community must participate in achieving this end, thus realising a providential and historical necessity. Because every individual possesses their own possible intellect, it is necessary for all individuals to contribute to a collective effort to attain earthly knowledge and the consequent achievement of one of the two ends and the happiness that results from it.

### 4. Between Autobiography and Palinode: The Motif of *Ingegno*

When examining the presence of Averroistic themes in the *Commedia*, the respect and homage that Dante pays to this stream of thought by including Averroes among the *magnanimi* in Limbo is first identified. Later, Dante brings Averroism into the context of his own time, which was not limited to philosophy but was common and widespread, touching on worldly matters and leading to the heretical belief in the mortality of the individual soul. This is evident in the presence/absence of Guido Cavalcanti among the heretics in *Inferno* X. This progression can be seen as portraying 'una 'istoria' dell'averroismo latino, considerato dalla sua grandezza originaria (*Inferno* IV) al pericolo presente (*Inferno* X) al suo attuale riscatto con Sigieri (*Paradiso* X)'[14], passing by the moment in *Purgatorio* XXV where Dante takes a clear stance against the unity of the intellect, still preserving the greatness of Averroes' figure ('*più savio di te fé già errante*', 'it led one wiser than you are to err', 63).

While it is reasonable to agree with Veglia's assessment that there is something highly autobiographical for Dante in these moments, which played a significant role in his formation and philosophical reflection, his study of Aristotle, and the correct use of human reason in the epistemological tension towards the divine, I do not entirely believe that this autobiographical character necessarily amounts to a palinode. Indeed, this would imply that Dante had fully adopted Averroistic positions at some point in his works, which does not seem likely based on the brief overview provided above and thorough recent scholarship on this topic.

Dante's earlier works leading up to the *Commedia* certainly drew from Averroistic thought. In the poem, Dante returns to reflect on human *ingegno* with greater maturity and from a different perspective. However, rather than retracting a position that was previously adopted, he places his quest for truth beyond it, recognises its value, condemns theories that oppose the immortality of the soul, then admonishes himself—also through Beatrice's reprimand[15]—and reminds the reader how to properly use human reason and intellectual curiosity. He also condemns and distances himself from all manifestations of excessive confidence in the individual's ability to achieve earthly happiness on their own[16]. Furthermore, Dante emphasised the specific error of Averroism in *Purgatorio*, which led a great thinker to consider the possible intellect as unique and universal rather than individual. Finally, he included this manifestation of the nobility of philosophical reflection within God's plan for eternal salvation under the right and inscrutable conditions with Siger of Brabant.

Certainly, the autobiographical element is clearly present; for this reason, the theme recurs throughout the poem. Dante's insistence on returning to the motif of *ingegno* and how it is necessary to restrain its misdirected enthusiasm, which could only lead to the loss of this very good and its potential for eternal beatitude, stems from this autobiographical

character. In distancing himself from these elements, the poet tempers certain aspects of himself and his previous poetics. This is not necessarily a retraction, but a way of setting different positions in the right perspective and correctly distributing value between reason and grace. Dante is undoubtedly marked by his encounters with an extremely radical and fatal 'disdain' through his friend Cavalcanti, whose presence is detectable in *Inferno*, as well as the pride and intellectual restlessness of classical culture and its representatives, such as Ulysses. At the same time, he is influenced by the harmony, balance, humility, and intellectual honesty that are typical of a thinker such as Siger, who earned his place in Paradise during his lifetime.

While it may not be a full palinode or retraction, and Dante's intellectual journey is not necessarily linear or progressive, the *Commedia* does represent a significant rupture from everything that came before in Dante's works. It implies the realisation that, after probing the limits of human speculation and reaching its extreme boundaries, the only way to transcend them is not through philosophical *continuatio* but a discontinuous leap that is made possible only through grace. This leap is also the only one that allows the human intellect to open up to the divine rather than close itself within its own limits.

In this sense, a point of connection is provided by the opening of *Paradiso* XI. Right after the atmospherically pacified and eclectically harmonious scene that transcends human and philosophical discord in the closure of *Paradiso* X, Dante inveighs against *l'insensata cura dei mortali* [senseless cares of mortals] as follows:

> O insensata cura de' mortali,
>
> quanto son difettivi silogismi
>
> quei che ti fanno in basso batter l'ali![17]

This is a point of connection because the deductive speculation based on syllogisms, which was praised in the previous canto as capable of attaining truth (Siger '*silogizzò invidïosi veri*' ['demonstrated truths that earned him envy'])[18], is once again cautioned against here if it were to produce 'defective syllogisms'. In both cases, the reasoning of Aristotelian and scholastic logic is at stake, the same philosophy that finds place in paradise, if turned exclusively towards the lower immanent life and its goods, falls into error, falsehood, and damnation. Additionally, Siger's fate is connected here to that of Ulysses, which is vaguely echoed by the expression '*in basso batter l'ali*' [bring your wings to flight so low]. Indeed, *Inferno* XXVI opens with the famous invective against Florence:

> Godi, Fiorenza, poi che se' sì grande,
>
> che per mare e per terra batti l'ali,
>
> e per lo 'nferno tuo nome si spande![19]

The greatness known by Florence is only a horizontal and immanent greatness, a flapping of wings over sea and land that is a flapping of wings below, so low as to reach and populate Hell extensively. A few tercets later, the reader encounters Dante's self-admonition to his own intellect, which he urges not to roam down paths ungoverned by virtue.[20]

Fundamentally, what interests Dante the most with regard to examining the presence of an Averroist philosopher in Paradise is the philosophical theme of the proper application of human intellect (*ingegno*). This is evident by comparing the presence of the motif of *ingegno* in the invocations at the beginning of each cantica and its progression from *Inferno* X to *Purgatorio* XI and *Paradiso* X.

First, the poet begins to recall the journey by invoking his '*alto ingegno*' [high genius] and the nobility of his '*mente*' ([memory], *Inferno* II, 7–9). Then, among the heretics, when questioned by Cavalcante about whether his privilege is '*per altezza d'ingegno*' ['high intellect'], Dante replies '*da me stesso non vegno*' (['My own powers have not brought me'], *Inferno* X, 58–61). Upon reaching the second realm and drawing a parallel between Ulysses' and his own voyage this time in the humility of grace, Dante inaugurates the second cantica by speaking of '*la navicella del mio ingegno*' (['my talent's little vessel'], *Purgatorio* I, 2). Later,

in Purgatory, he asks God for His peace to come towards mankind and acknowledges that the reverse process is impossible even '*con tutto nostro ingegno*' ([though we summon all our force], *Purgatorio* XI, 7–9). Finally, at the entrance to the paradisiacal realm, Dante stages the dissolution of '*nostro intelletto*' (['our intellect'], *Paradiso* I, 7–9) as he approaches to merge with the fullness of his desire. In this case, desire is understood in all of its full potential since it is outside the realm of reason, unlike in the *Convivio*. During this process, cognitive faculties lack the power to preserve a rational and gnoseological trace of this experience in memory.

Following this moment in *Paradiso* II, the poet stages a type of '*anti-orazione*' [counter-prayer] compared to Ulysses' one[21]. Among the evident parallelisms, the most striking is the correspondence between '*considerate la vostra semenza*' ['consider well the seed that gave you birth'] and '*tornate a riveder li vostri liti*' [turn back to see your shores again], which have opposite functions; the former lures listeners into an enterprise that is greater than their abilities and destined for failure, however noble, and the latter encourages readers to ensure that they have not only the *altezza d'ingegno* [high intellect] but also the faith and grace needed to enable such an experience. Thus, while Ulysses is a deceitful counsellor who urges his companions to follow him into damnation with his skilled words and leadership, Dante is a poet of truth who safeguards and protects readers from flying higher than their own strength allows with his sacred words. He urges them not to set sail on an eager Ulysses-like *navigatio ingenii* precisely to prevent them from becoming lost, not by going with him, but by losing his guidance:

> O voi che siete in piccioletta barca,
>
> desiderosi d'ascoltar, seguiti
>
> dietro al mio legno che cantando varca,
>
> tornate a riveder li vostri liti:
>
> non vi mettete in pelago, ché forse,
>
> perdendo me, rimarreste smarriti[22].

Nevertheless, Dante authorises a certain portion of readers who have been able to accompany philosophical knowledge with theology and who, therefore, find themselves in the position to follow him in his endeavour to sail the '*alto sale*' ['deep salt-sea'] up to paradise, so different from Ulysses' '*alto mare aperto*' ['open sea']:

> Voialtri pochi che drizzaste il collo
>
> per tempo al pan de li angeli, del quale
>
> vivesi qui ma non sen vien satollo,
>
> metter potete ben per l'alto sale
>
> vostro navigio, servando mio solco
>
> dinanzi a l'acqua che ritorna equale.[23]

Similar to *Paradiso* I, in the Heaven of the Sun among the Wise Spirits, Dante draws attention once more to the fact that '*perch'io lo 'ngegno e l'arte e l'uso chiami*' ('Though I should call on talent, craft, and practice', *Paradiso* X, 43); he cannot verbally describe what he experiences in a way that would allow the reader to imagine it. Instead, the reader must believe him and desire to be called to live the same experience[24]. A few verses later, there is another mention of *tanta altezza* ('such heights', 47); this time, however, it is not the height of intellect that is advocated for but the extraordinariness of the paradisiacal vision, which renders our fantasies so low.

## 5. Dante and Siger: From Error to Truth

Critics have emphasised the importance of investigating the specificity of the relationship between Dante and Siger of Brabant, as each declination of Averroistic thought was personal. This enables us to better explore Siger's presence in Paradise, which has been considered 'suggestive' and 'potentially problematic'[25] due to his summons to respond

to the French inquisitor Simon du Val about his errors in 1276 (an accusation of heresy from which he was later acquitted due to lack of evidence) and the 30 articles censured by Tempier in 1277, which were related to his opinions.

Although we are unaware of what Dante might have read from Siger, what the poet presents to us in Paradise is 'the portrait of an intense, troubled and powerful intellect'[26], a philosopher and master of the arts who was able to reach uncomfortable truths, evidently not only for the Church but also for himself.

Based on the role that Thomas plays in introducing him to Dante in the Heaven of the Sun, we can imagine that the poet was familiar with the debate between the two regarding Thomas's *De unitate intellectus contra Averroistas* (1270) and Siger's response in *De anima intellectiva* (1272–1273). In the latter work, Siger moderated his views on the unique intellect and self-sufficiency of philosophy, respecting theological knowledge.

Moreover, Dante was likely aware of Siger's violent death, given the testimony found in *Il Fiore*[27], even without attributing it directly to Dante. Indeed, *Il Fiore* recalls Siger's assassination, which occurred between 1281 and 1284 at the hands of an insane secretary in Orvieto. Siger had gone there, likely to negotiate matters related to his previous accusation of heresy, which was later dropped (Bianchi 2014a, p. 88). In this work, Siger's death is presented as a 'political' murder, which indicates the killing of an inconvenient philosopher by an Inquisition hitman (Falzone 2016, p. 10).

Falzone suggested that Dante might have known only a few significant biographical details about Siger due to the vagueness of the information presented in *Paradiso* X (133–138). Nevertheless, even if the accusation of heresy is not taken into account, the scholar argued that Dante must have been aware of Siger's tragic death in the papal curia in Orvieto[28]. These details could have allowed Dante to portray Siger as another model of justice besieged by power, a prevalent theme in his *Commedia*. However, as Bianchi highlighted, Siger was not only a legendary figure during Dante's years of philosophical education but also a respected philosopher. Indeed, the texts of the Parisian masters of arts had spread to Italy quite early (Bianchi 2014b, pp. 80–87).

Therefore, it has been argued that Siger in *Paradiso* ultimately represents the harmony between the truths of faith and reason in the celestial realm, which welcomes and reconciles the Wise Spirits through the integration of their knowledge in different fields of expertise. Thomas and Siger are representatives of the Faculties of Theology and Arts, respectively. The fact that the former praises the latter signifies the transcendent resolution of academic disputes and, ultimately, the free nobility of philosophical speculation in all of its facets. It is also noteworthy that Thomas Aquinas is positioned between his teacher Albertus Magnus and his adversary, which conveys through spatial arrangement[29] that knowledge is built on dialectical tension in learning and that all of its declinations exist on the same plane, just as the three souls represent some of the greatest proponents of scholastic Aristotelianism in cultural synergy.

The presence of an Averroist in *Paradiso* certainly raises questions about the contrast between the eternal damnation of Averroes (who was wiser than Dante but went astray), the fate of heretics, and Siger's *luce etterna* [everlasting light] because of his *invidïosi veri* [truths that earned him envy]. Siger cannot be considered a heretic by Dante, nor does the poet place him as a spirit purging the sin of pride. From this it is possible to infer that, according to Dante, Siger had already achieved salvation on Earth, where he had been on the right path.

It is essential to remember[30] that, during the Middle Ages, the term 'heresy' had a very different meaning from 'error'. A heretic was an individual who stubbornly maintained their positions even when they contradicted truths of faith. For a medieval thinker such as Dante, a wise person was capable of facing error, correcting themselves, and overcoming it by making distinctions, which was characteristic of scholastic logic. Thus, it can be argued that, according to Dante, Siger had such a wise response to Averroistic error; he did not reject the truth but rather embraced it[31].

The potentially heterodox issue is that, within the framework of his teaching, Siger, in reference to the Long Commentary on the third book of *De Anima*, argued that man is capable of understanding separate substances—although he usually taught that man can only know God through His effects—and went so far as to claim that 'a man very experienced in philosophy' could contemplate 'the essence of the First' in life[32]. This thesis was censured by Tempier in Article 36.

Naturally, this position was antithetical to what Thomas Aquinas argued. Thomas believed that it was possible for man to understand that God is—quia est—but not what God is—quid est—through philosophical knowledge. He believed that this should lead man, in his desire to know the essence of God, to seek complete happiness in eternal life and the transcendent as a viator towards the spiritual goal, which is precisely the theme of Dante's poem.

However, as far as it is known, some of the theses condemned by Tempier also originated from Thomas's teachings. Thus, this condemnation affected not only the Faculty of Arts but also that of Theology. Thomas had been criticised and censured at Oxford in 1277 and 1286; in 1282, the Franciscan order restricted the use of the *Summa Theologiae*, which was considered dangerous for simple readers. It was only in the early decades of the 14th century that Thomas began to be identified as a theologian of an invincible orthodoxy that was opposed to Averroes, and his reputation remained controversial even until the writing of *Paradiso*. Among the Franciscans, he was even perceived by some as an inspirer and accomplice of the radical Aristotelians (Bianchi 2014a, pp. 90–91).

Therefore, the positions of Thomas and Siger should not be overly polarised or, rather, the intellectual tension between the two should not be viewed as resolvable into the dichotomy of the 'champion of orthodoxy' and the heretical Averroist. Similarly, Dante and Siger's positions should not be polarised, like those of the Faculties of Arts and Theology, and it is important to pay attention to these delicate relationships in their complexity and to which influence Dante received from them. Indeed, since Thomas was also a target of Tempier, Siger and Thomas were, in a way, on the same terrain, although they occupied very distant parts of it.

## 6. Dante as the Last Persecuted Intellectual

It has been observed that Dante closely identified with the only two philosophers in the strict sense present in the 10th canto: Siger and Severinus Boethius (who, for Dante, was primarily the author of *De consolatione philosophiae*). These are also the figures on whose deaths Dante lingered over the most, and whom he considered victims of injustice[33].

The condition of contemporary philosophers was indeed perceived as under attack, and knowledge was persecuted (from which Dante's anti-academic and anti-ecclesiastical attitude), as it had been for ancient philosophers such as Boethius, who had suffered exile ('*da martiro/e da essilio venne a questa pace*' [he came/from martyrdom and exile to this peace])[34] and in whom Dante certainly saw something of himself. He placed himself within this genealogy of persecuted, oppressed, unjustly condemned, and expelled sages and entered the circle of ancient sages and philosophers already in Limbo to paradigmatically share the history of suffering that extended from the past to his time and ultimately culminated in himself.

Additionally, in Limbo, the verse '*fui sesto tra cotanto senno*' ['I was the sixth among such intellects'][35] metaphorically recalls the fact that Dante, in the past of the *Convivio,* was animated by the same desire for knowledge as these philosophers. He participated in their earthly mission and the idea of human happiness as independent and self-sufficient. Indeed, Dante's journey is not only diachronic through his works but also textually synchronic within the *Commedia* itself, as it ascends from Hell to Purgatory to Paradise.

Dante was undoubtedly interested in the fates of Boethius and Siger for this reason, and it is not insignificant that he placed them in Paradise to enjoy eternal bliss at the highest level of wisdom. Dante viewed the fact that their knowledge was rejected on Earth with pride and claimed it for himself[36]. Regarding Siger, it is difficult to determine whether

Dante considered his death a 'political' murder and thus perceived him as a martyr of philosophy. However, Dante did mention his death; even through *Il Fiore*, he likely knew that it was a violent death at the papal court but remained vague and obscure about it. Nevertheless, Dante unquestionably felt a parallel, which emerges in his treatment of the two philosophers in the canto.

However, beyond the biographical element of the persecution of intellectual research, it is legitimate to wonder how Dante could feel this consonance with a philosopher who held Averroistic positions, which were unacceptable to a Christian believer. To investigate this question more thoroughly, it is important to bear in mind that medieval culture was acutely aware of the distinction between the object and method of philosophy and theology. When commenting on the works of Aristotle, Siger and the masters of the arts distinguished between 'expounding' (*recitare*) and 'asserting' (*asserere*) the philosopher's ideas (Bianchi 2014a, p. 101). Thus, they made a distinction between the attitude of the philosopher and that of the believer. The latter involves not only another type of intellectual engagement, but also a spiritual commitment. Accordingly, what can be considered true by way of natural reason and thus philosophically true is different from what is true in faith. The epistemological rule that medieval thinkers followed (which came from Aristotle himself) was that every specialist in a science must stay within the boundaries of their own science.

Bianchi indicated that this could lead to two different attitudes. On the one hand, this states the autonomy of rational disciplines against the subservice of philosophy to theology. On the other hand, it sets clear limits on these sciences that they must recognise and not exceed. Thus, the masters of the arts were well aware that their conclusions were valid within the scope of a certain field of knowledge and under certain premises. However, their conclusions did not hold as truths in every context because the supernatural could suspend or alter these principles[37].

Siger also adopted this profound recognition, as he believed that Aristotle's proposal of the eternity of the world in his treatises on physics, cosmology, and metaphysics was valid only within these disciplines. In fact, in the *Quaestiones in tertium de anima*, in which he argues for the unity of the possible intellect, he claims that all beings are produced by the First Cause and that their duration depends on the form of its will[38]. Since this will is inscrutable, it is impossible to rationally determine whether it has given the world finite or infinite duration. Therefore, Siger had an agnostic stance on these matters that was inspired by Thomas. In the same work, Siger presents a distinction between Aristotle and Augustine's conceptions of the eternity of the intellect. However, the conclusion that he reaches is what mattered to and fascinated Dante. On the one hand, the philosopher admitted that, following Aristotle's reasoning, one should support the eternity of the intellect. On the other hand, he argued that this is not necessarily true, precisely because of the unfathomable omnipotence of the divine.

Therefore, given that Aristotle and Augustine's positions were opposed and that it is impossible to agree with both, Siger did not establish a hierarchy. He was uninterested in determining the superiority of one position over the other according to a criterion of 'true' and 'false'. He respected both of them equally, although he leaned towards the probability of the first within a dialectical argument that contemplated the non-necessity of the philosophical conclusion.

Siger's conclusion about these issues was that, if the principles that gave rise to them are denied, these conclusions can also be denied. The principles are 'probable' but not 'necessary'[39]. What resulted from this attitude towards philosophy was that, as an autonomous but limited field of knowledge, it had legitimacy even when it arrived at conclusions that differed from the truths of faith. Indeed, this did not contradict those truths precisely because of the distinction that philosophers made in their preliminary studies. Philosophers have a natural perspective on the world governed by causes and effects, as illustrated by Aristotle, while believers go beyond this and the natural order of the world and thus have a different perspective on it.

What Siger asserts in *De anima intellectiva* is fundamental. In this work, the philosopher recalls that his task is not to investigate revealed truths about the soul. Rather than seeking absolute truth, the philosopher's task is to uncover a truth that is relative and circumscribed by the '*intentionem philosophorum* [...] *magis quam veritatem*'[40]. This represents a sort of pre-humanistic attitude and rigor in the philological pursuit of philosophical truth and certainly not divine truth.

As Nardi also recalled (Cf. Nardi 1960c), Averroism is first and foremost an interpretation of Aristotle's doctrine on some significant issues. Accepting this interpretation did not necessarily imply acceptance of Aristotelian philosophy. Rather, it indicated recognition that there was opposition between the teachings of philosophers and those of faith regarding these issues.

This helps us to understand Dante's choice to place Siger in Paradise and his praise of his philosophical mission. In a place such as Paradise, every form of knowledge, including philosophy, contributes to creating a complex and plural picture of truth rather than undermining its unity.

## 7. Aristotle Can Err: The Importance of Questioning Authority

Another important level of Siger's positioning within the dialectical tension between reason and faith is what he illustrated regarding authority[41]. I believe that an important aspect of Dante's relationship with this Averroist philosopher lies here.

As mentioned earlier, Dante certainly admired the reasoning that was also present in *Quaestiones in Metaphysicam*[42], which suggests that Aristotle's intention should be honestly investigated and that the path of reasoning in his works should be thoroughly explored, even when it leads to conclusions that clash with the Church's doctrine. This does not mean that reason can achieve the contemplation of what lies beyond its domain and transcends it in terms of object, method, and penetrability.

However, the key point of the argument, which must have deeply struck Dante, is that, despite the greatness of the philosopher, his knowledge is subject to error, as shown in Dante's verse '*più savio di te fe' già errante*'. From this follows the folly of thinking that philosophical conclusions can call into question universal truths of faith. Brazeau succinctly summarised this concept as follows:

> What stands out in the above passage is Siger's insistence that the findings of philosophy cannot simply be collapsed into the truths of faith, even if the philosopher does not know how to refute what appears to be a contradictory truth[43].

Therefore, it can be asserted that the seriousness and intellectual honesty of Siger's investigative method, such as his commitment to knowledge and the intention of the philosopher, are all elements that Dante deemed worthy of paradise, especially in the figure of a persecuted philosopher whose knowledge is under threat. A strongly Dantean element is precisely the questioning of authority present in Siger, both as a concept and as a questioning of what is asserted by Aristotelian philosophy, on which Dante based the universe of his sacred poem. The concept of overcoming authority was fundamental for the poet, who built the *Commedia* on a constant dynamic of correction—first towards Virgil's guide, a dynamic that provides the premise for the leap that he realises with the composition of such a work.

Brazeau argued that admitting Aristotle's potential fallibility would disturb Dante's notion of the two beatitudes, as it is based on the philosopher's statement that man is composed of a corruptible body and an incorruptible soul in Book II of *De Anima*. By contrast, I believe that the fallibility advocated by Siger, being theoretically and therefore only potentially in line, does not disrupt this Dantean concept. Dante maintains all of its foundations in Aristotle's assertion. On the contrary, I suggest that the admission of the possibility that even the greatest of authorities may err is not only one of the strong points in determining Siger's salvation, but also one of the main assumptions of Dante's poetic operation. It was the necessary presupposition for the poet to go beyond all previous

knowledge, to dare what had never been dared before in vernacular poetry, to present philosophical and theological truths together, to reveal eschatology to the widest audience without omitting the vision of the divine, and thus to establish his own new authority above previous ones.

The way in which Dante constructs his authority, not only as a poet but specifically as a scribe of the divine, radically involves the correction of errors made by the auctoritates who came before him. This process is so significant that it becomes the paradigm for his disciple–mentor relationship with Virgil. It revolves around the unveiling of a truth to which Dante has access but others before him did not. It is about completing a partial and fallible knowledge, such as the ancient pagan wisdom, despite its greatness and magnanimity.

This even goes beyond that: Dante also disrupts certain theological doctrines. Some examples can be provided by his representation of Limbo, the salvation of pagans such as Cato or Rifeus (or Averroists such as Siger), the salvation of some emperors under exceptional circumstances (Trajan), but not necessarily popes, and the salvation of sinners and the excommunicated (Manfred). This not only demonstrates that the greatness of philosophers can be fallible but also the ecclesiastical attitude that condemns without appeal and does not always align with the inscrutability of the divine will.

Similarly, in the salvation of Siger, Dante symbolically distanced himself from not only Tempier's ecclesiastical attitude and condemnations, but also from the institutional academic perspective of divisive disputes within knowledge. This perspective aimed to reduce everything to an unattainable linearity and coherence in understanding. The poet condemned sterile disagreement and the closed polarisation of viewpoints in various disciplines rather than the embrace of dialectical debate that advances knowledge through distinctions. The contrary reflects the involution of science into the desire to establish the superiority and infallibility of one's own discipline at all costs, especially at the cost of becoming irrelevant to civic, political, and ethical life, both immanent and transcendent.

In this sense, an emblematic moment is when Dante, through Beatrice's words, denounces the confusion of truth that becomes equivocal when knowledge divides itself not to enrich, but from a craving for the supremacy of a single perspective, a singular interpretation to be absolutised. However, the poet states that this condemnation in heaven is less severe than the utilitarian distortion of the sacred Scriptures:

> Ma perché 'n terra per le vostre scole
>
> si legge che l'angelica natura
>
> è tal, che 'ntende e si ricorda e vole,
>
> ancor dirò, perché tu veggi pura
>
> la verità che là giù si confonde,
>
> equivocando in sì fatta lettura.
>
> [. . .]
>
> Voi non andate giù per un sentiero
>
> filosofando: tanto vi trasporta
>
> l'amor de l'apparenza e 'l suo pensiero![44]

Therefore, it is the three souls of Albert, Thomas, and Siger, each with diverse approaches to attaining and contributing to knowledge, that realise this ideal of harmonious trans-academic concord to readers.

As far as this concord is concerned, Bertolacci discussed a 'prospective vision' of the '*filosofica famiglia*' [philosophic family, *Inferno* IV, 132] in Limbo and viewed harmony as reflected in the circular arrangement of philosophers—around the historical centrality of Aristotle—as a foreshadowing for Paradise; in this way, the concentric description of philosophers and scientists prefigures the two crowns of the blessed theologians depicted in the Heaven of the Sun in *Paradiso* X–XIV[45]. Thus, the magnanimous souls in Limbo, especially the philosophers at the apex position, anticipates those saved souls who have combined magnanimity with humility and grace.

With regard to authority in Limbo and among the Wise Spirits, it is possible to agree with Brazeau's argument that this question is central in the Heaven of the Sun, especially because Dante explicitly drew the reader's attention to his own text and his *auctoritas*, which he explicitly presented as that of a prophet:

Or ti riman, lettor, sovra 'l tuo banco,

dietro pensando a ciò che si preliba,

s'esser vuoi lieto assai prima che stanco.

Messo t'ho innanzi: omai per te ti ciba;

ché a sé torce tutta la mia cura

quella materia ond'io son fatto scriba[46].

Now, it is the poet himself who holds a higher authority that must be followed, even if it is beyond our understanding, precisely because it concerns a sacred matter. He conveys this message in the same canto in which he praises the Averroist philosopher who was able to recognise the potential fallibility of previous *auctoritates*. In some ways, Siger paved the way for Dante, much like Cavalcanti did in poetry, in a significant parallel between *Inferno* X and *Paradiso* X.

Brazeau suggested that Dante responds to Siger's discourse on the fallibility of Aristotle by reaffirming his authority (in the *Convivio* and the *Monarchia*) and ultimately naturalising his own authority as a philosopher, poet, and theologian through rhetorical strategies in *Paradiso* X[47]. On the other hand, I suggest that Dante did not have the necessity to reaffirm Aristotle's authority as Siger had only questioned it in logical potential terms in order to equally respect the authority of theological knowledge, which he openly contradicted when following Aristotle's reasoning about natural philosophy and the nature of the soul and the world. Moreover, I propose that Dante may have agreed with Siger on this questioning of authority, which he himself continually practiced in the poem to establish his own innovative authority. Indeed, there is no explicit defence of Aristotle as absolutely infallible in the poem. In this regard, it is sufficient to examine the ambiguous treatment that Dante reserved for the beloved guide, who is precisely the embodiment of this Aristotelian knowledge and authority.

It is also possible to reflect on whether Dante applied this concept of fallibility to his own authority, but the answer lies in the fact that he presented himself as a scribe, the writer of a sacred poem whose role is merely to '*notare*' [take note][48] the askesis to God and the eschatological realities. His authority is not only that of a philosopher but also that of a theologian, which is why it is infallible[49].

Therefore, it could be precisely this profession of *humilitas*[50]—rather than a conversion to Thomism and its alleged orthodoxy, as has been argued[51]—which removed the risk of pride and philosophical hubris and earned Siger his salvation.

Indeed, as it is impossible to determine which of the Averroist philosopher's texts Dante read, Veglia discussed a 'spiritual consonance'[52] between Dante and Siger, regardless of whether the former read Siger's *Quaestiones*, in which Siger expounds that the powers of reason alone are fallible in comparison to the divine omnipotence. Consequently, they are insufficient for achieving perfect beatitude in God. Such consonance would have been mediated by the *Quaestiones* or Dante's frequentation of an Averroistic environment in Bologna, in which he would have conversed with those who had read and commented on this text in philosophical disputes. Thus, Veglia concluded,

Dalla permanente difficoltà di accertare, senza forzature, quali "testi" nutrissero oggettivamente il pensiero dantesco, possiamo trascorrere alla coincidenza stringente dei "contesti" che, dalla *Commedia*, ci conducono ad aree culturali conosciute *con certezza* da Dante. [. . .] Se anche non troveremo mai il testo preciso, la cosiddetta "fonte" del filosofo brabantino nell'universo filosofico di Dante, non per questo dubiteremo che quel filosofo, per determinati problemi circoscritti da determinati contesti, fosse presente a Dante e caro al suo itinerario spirituale[53].

To further support the consonance between the conceptions of the two, it is interesting to recall Nardi's mention that the Averroistic current of Siger, even when advocating for the thesis of the unity of the intellect, aimed to reconcile it with the view of the unique intellect that individualises in men and becomes their substantial form, as it is by its nature made to be united to the human organism already endowed with vegetative-sensory life, with a substantial bond, to constitute it in its being of man (Nardi 1960b, p. 220). Nardi argued that Siger specifies this in the treatise *De intellectu* in response to Thomas and states that the single intellect for all men is ordered to inform their 'cogitative' faculty, thus constituting the composite rational soul that is truly the substantial form of man (Nardi 1960d, p. 161). For Dante, this was crucial in maintaining the individual character, not only the universal character, of knowledge, the participation of the individual in thought, and thus the individuality of consciousness, a point that Averroes' doctrine threatened. This represents a synthesis between individuality and universality, '*quell'uno nei molti, fuori dei quali l'uno non ha più senso e svanisce nel vuoto dell'irrealtà*' [That one within the many, outside of which the one loses its meaning and fades into the void of unreality] (Nardi 1960b, p. 222).

## 8. Ouverture

*Action and Reaction: Antithetical Destinies*

In Dante's universe, it is not only individuals who are judged by their singularity but also ideas and systems of thought. The validity of a certain perspective is assessed, within which each individual may make different choices, including whether to embrace the grace that can revolutionise their destiny. Barański expressed the following view of heresies:

> What is striking about the poet's reaction to such errors of doctrine is that he refrained from attacking the individuals who held them in order to focus attention on the ideas themselves. It was the heresies rather than the heretics that needed combatting, since it was the former that inflicted damage on the faith[54].

Starting from this point, it can be suggested that there is a kind of reasoning with antithetical and anticipatory pairs in Dante. For instance, one can consider the case of Virgil and Statius, which Veglia makes proportionally correspond to the relationship between Guido Cavalcanti and Dante[55]. The dialectic is one of overcoming, of anticipatory announcement, as is for the paradigmatic pair of John the Baptist and Christ.

It was Veglia who affirmed that Dante, in his poetic turn, took inspiration from Guinizzelli but made Cavalcanti his precursor and St. John and overcame him through a dialectic of humility and magnanimity.

It would be interesting to see whether this is also applicable to the treatment of Siger. Maria Corti identified a *gradatio*, which would be a backward movement in exorcising the ghost of radical Aristotelianism: from the clear rejection in the canto of the heretics to the fascinating symbolic representation of Ulysses to a reasoned revaluation of the last Siger in *Paradiso*, always due to supernatural causes (Corti 1982, p. 97).

More generally, she discussed a *reductio ad unguem,* an ethical–metaphysical type of dialectical and antinomic culture of the time, referring to *Inferno* X and *Inferno* XXVI. According to the scholar, Dante came to terms with his youthful enthusiasm for every form of knowledge in these cantos and was wrapped in the fascination and respect that the censorship of these scholars' intellectual curiosity (with Tempier's condemnation in the first place) exerted on writers of Dante's generation, even if they had changed their views[56].

Corti sees in Siger, as a representative of the epistemological autonomy of secular philosophical thought, a Faustian character '*sempre pronto a perdersi, ma alla fine salvo*' [Always ready to get lost, but in the end, safe][57] in a universe in which it is consistent that he is introduced to Dante by Thomas Aquinas, as he would have brought him back to orthodoxy[58].

However, Corti's concept of *gradatio* does not seem too convincing, especially because Dante cannot be strictly considered an Averroist even when he exalts the trajectory of philosophical knowledge to the fullest in the *Convivio*, as he always maintains it within the

bounds of human potentialities. Instead, I suggest that, more than a linear progression, Dante simultaneously presented parallel destinies in the universe of his poem. In a certain sense, these parallel destinies are equal and opposite: equal in that they are based on the same system of ideas or area of knowledge, but radically opposite in terms of their outcome in the afterlife. The shift would be operated by the antithesis between magnanimity and humility, wisdom and grace.

This reasoning could apply to various pairs in a sort of binary system that intertwine multiple times. As far as the area of poetry is concerned, I already mentioned the pair of Virgil and Statius, who were united by the heights of poetry, or rather, the former was superior to the latter. However, Statius was saved precisely because he overturned the dialectic with humility thanks to Virgil. Virgil, the torchbearer, bears the traits of John the Baptist[59], while Statius is presented in the canto as Christ when he appears to the disciples of Emmaus[60].

Similarly, with regard to politics, another pair is represented by Frederick II and Manfred. The latter surprises the reader with his presence in *Antipurgatorio*, which is also a way for Dante to condemn the attitude of Pope Clement IV, who had him tried for heresy and excommunicated. Manfred's salvation is made possible by his humility, which leads to repentance and thus salvation; this contrasts with Frederick II's pride, which lands him among the heretics.

Therefore, with regard to philosophy and the discussion of Averroism more specifically, the pairs that assume particular significance are those that give rise to an interesting chiasmus: Guido–Siger and Ulysses–Dante. If Guido's damnation is outlined in *Inferno* X, it corresponds in a reversed manner to Siger's eternal light in *Paradiso* X. In a specular way, it is known that Dante presents himself in the role of a new Ulysses who crosses the boundaries of the known and ventures into uncharted waters, but with a significant difference. This time, the poet is a scribe who has been entrusted with a mission and is therefore authorised to undertake such an enterprise. Then, the shipwreck is Ulysses' but certainly not Dante's.

What is interesting is the chiasmus between the two pairs Guido–Dante and Ulysses–Siger. Regarding the first pair, the prefigurative relationship between the two poets exposed by Dante in *Purgatorio* XI is evident. As for the second pair, it is interesting to draw on the correspondences highlighted by Corti (Cf. Corti 1993) and Freccero (Cf. Freccero 1988), among others. Their works illustrate how metaphors related to navigation were applied to the concept of human reason as *nauta navis* to signify the epistemological hubris of man. This is in close correlation between venturing beyond what is licit and the Averroistic philosophical project.

## 9. Conclusions

What clearly emerges from this study is Dante's 'holistic' attitude and approach to truth. Since the *Commedia* aims to deliver the revealed realities, the eschatological framework of the world, poetry becomes the place in which every form of knowledge converges in its most varied manifestations.

It is indeed only through a supernatural combination of partial views of reality—fragments of knowledge as they are spread on Earth—that Dante can access the truth in its entirety and its seemingly contradictory, non-linear, and destabilising totality, which ultimately expresses its complex and authentic character.

Thus, at a more structural level in the poem, the presence of Siger in Paradise is a manifest expression of the manifold within the one, it carries the harmonious *contrappunto* of a philosophical thought that, however discordant it may have appeared on Earth, professed the rigorous deepening of philosophy insofar as it can convey truth to us and promote scientific inquiry. This knowledge appropriated a well-defined domain, method, and subject, and acted with the utmost seriousness in its study without being intimidated by the problematic conclusions it could reach or the authorities it could disturb. As stated in Chiavacci Leonardi's commentary, Siger's *veri* were refused by the ecclesiastic powers

precisely because, in exalting the human mind, they subtracted the thinker from any kind of control. This is the liberty that Dante celebrated in the canto—the same liberty on which he founded his entire poem.

This type of knowledge and rigour is a constitutive part not only of Dante's education, but also of his works and the poem in particular, which is traversed by the theme of Averroism. That it finds space and is integrated into the *Commedia* is then fundamental in relation to the sacred status that Dante attributes to his own work, and to his figure as invested with the mission of unfolding the divine and making the entire present and future human community participate in his journey.

The truth with which Dante deals, so different from the philosophical and scientific truth at the heart of the Averroistic tradition and the *Convivio*, indeed demands precisely that knowledge perceived as antithetical in medieval disputes be reconciled and presented in a perspective that has a heavenly elevated viewpoint on the realities of the world. Thus, examining them from above not only relativises and places them in the correct position within the design of divine truth, but also preserves all of their human and historical character. Let us just think of the delicacy of the cone of light that opens on the 'Vico degli Strami' on Earth, in words spoken by a soul inhabiting the so-distant, blinding Paradise.

**Funding:** This research received no external funding.

**Institutional Review Board Statement:** Not applicable.

**Informed Consent Statement:** Not applicable.

**Data Availability Statement:** Data is contained within the article.

**Conflicts of Interest:** The authors declare no conflict of interest.

## Notes

1. Bruno Nardi called this 'panpsichismo' [panpsychism] in (Nardi 1960b, p. 209).
2. Bianchi, *op. cit.*, pp. 94–95.
3. Barański, *op. cit.*, p. 292.
4. Cf. Bianchi, *op. cit.*
5. *Ivi*, pp. 77–78.
6. *Convivio* III, viii, 15; and IV, xxii, 13.
7. Cf. Nardi (1960a) and Barański, *op. cit.*
8. I return to the Ulysses–Siger pair later in this paper.
9. Barański, *op. cit.*, p. 302. This should be put in relation to the *Epistola XIII a Cangrande della Scala*, its attribution and the dialectic between '*allegoria dei poeti*' [allegory of the poets] and '*allegoria dei teologi*' [allegory of the theologians]. Indeed, the author of the letter openly states that the poem should be read as an '*allegoria dei teologi*', that is true also at its literal level.
10. *Inferno* X, 15. All the verses' translations are from Mandelbaum.
11. Barański, *op. cit.*, p. 326.
12. Bianchi, *op. cit.*, pp. 79–80.
13. This theory was dismantled as legend in (Van Steenberghen 1974).
14. Veglia (2000, pp. 73–74). 'A 'history' of Latin Averroism, considered from its original greatness (*Inferno* IV) to the present danger (*Inferno* X) to its current redemption with Siger (*Paradiso* X)', my translation.
15. *Purgatorio* XXXIII, 85–90.
16. Cf. Ulysses' episode, but also Virgil's admonition in *Purgatorio* III, 37–45, and the invocation for God's peace in *Purgatorio* XI, 7–9.
17. *Paradiso* XI, 1–3. 'O senseless cares of mortals, how deceiving/are syllogistic reasonings that bring/your wings to flight so low, to earthly things!'.
18. *Paradiso* X, 138.
19. *Inferno* XXVI, 1–3. 'Be joyous, Florence, you are great indeed/for over sea and land you beat your wings/through every part of Hell your name extends!'.
20. Cf. *Ivi*, 19–24.
21. Cf. *Inferno* XXVI, 112–120.

22    *Paradiso* II, 1–6. 'O you who are within your little bark/eager to listen, following behind/my ship that, singing, crosses to deep seas/turn back to see your shores again: do not/attempt to sail the seas I sail; you may/by losing sight of me, be left astray'.

23    *Ivi*, 10–15. 'You other few who turned your minds in time/unto the bread of angels, which provides/men here with life—but hungering for more—/you may indeed commit your vessel to/the deep salt-sea, keeping your course within my wake, ahead of where waves smooth again'.

24    In this canto, the address to the reader in verses 22–27, in which Dante refers to himself as a '*scriba*' ['scribe'], is also meaningful.

25    Barański, *op. cit.*, p. 314.

26    *Ivi*, p. 315.

27    Cf. *Il Fiore*, Edited by Gianfranco Contini (Milan: Mondadori, 1984), XCII. 9–11.

28    *Ivi*, p. 16.

29    For more on the disposition of souls in this canto, as anticipated by the disposition of the '*filosofica famiglia*' in Limbo, (see Bertolacci 2021).

30    Barański, *op. cit.*, pp. 304–305.

31    I will return to this point in the section 'Aristotle can err: the importance of questioning authority'.

32    Siger of Brabant (1981, p. 90): «Ita videtur quod homo multum expertus in philosophia, a causatis a Primo posset pervenire ad intellectum essentiae Primi».

33    *Ivi*, pp. 93–95.

34    *Paradiso* X, 128–129.

35    *Inferno* IV, 102.

36    '*L'essilio che m'è dato, onor mi tegno*', ['I count as an honour/the exile imposed on me'], *Tre donne*, 76, Princeton Dante Project's translation.

37    *Ibidem*.

38    *Ivi*, p. 102.

39    Bianchi mentions the following passages from (Pseudo-Siger of Brabant 1941, p. 200; Siger of Brabant 1972a, pp. 6–7; 1972b, p. 66; 1981, pp. 144–45; 1983, pp. 113–14, 413).

40    Cf. (Brazeau 2014, p. 112). Here, Brazeau reports Siger's quotation from *De anima intellectiva*, VII. 6–8.

41    Cf. Brazeau, *op. cit*.

42    Siger of Brabant (1983). III. 15, cited in Brazeau, p. 113.

43    Brazeau, *op. cit.*, p. 113.

44    *Paradiso* XXIX, 70–75; 85–87. 'But since on earth, throughout your schools, they teach/that it is in the nature of the angels/to understand, to recollect, to will/I shall say more, so that you may see clearly/the truth that, there below, has been confused/by teaching that is so ambiguous. [...] Below, you do not follow one sole path/as you philosophize—your love of show/and thought of it so carry you astray!'.

45    Cf. Bertolacci, *op. cit*.

46    *Paradiso* X, 22–27. 'Now, reader, do not leave your bench, but stay/to think on that of which you have foretaste;/you will have much delight before you tire./I have prepared your fare; now feed yourself/because that matter of which I am made/the scribe calls all my care unto itself'.

47    Brazeau, *op. cit.*, p. 125.

48    Cf. the poetic investiture in Beatrice word's in *Purgatorio* XXXIII, 52–54 using this biblical verb.

49    The reflection brings us again to the previously mentioned *Epistola XIII a Cangrande della Scala*.

50    Veglia, *op. cit.*, p. 98.

51    Cf. Falzone, *op. cit.*, p. 7 and (Bianchi 2014a, pp. 80–85).

52    Veglia, *op. cit.*, p. 102.

53    *Ibidem*. 'From the enduring difficulty of ascertaining, without forcing interpretations, which "texts" objectively influenced Dante's thought, we can move on to the compelling coincidence of the "contexts" that, from the Divine Comedy, lead us to cultural areas known with certainty by Dante. [...] Even if we may never find the precise text, the so-called "source" of the Brabantine philosopher in Dante's philosophical universe, we will not doubt that this philosopher, for certain issues circumscribed by specific contexts, was present in Dante's mind and dear to his spiritual journey', my translation.

54    Barański, *op. cit.*, p. 287.

55    Veglia, *op. cit.*, p. 93.

56    *Ivi*, p. 88.

57    *Ivi*, p. 98.

58    *Ivi*, p. 100.

59    A study of the relationship between the two figures is in progress, cf. (Portagnuolo 2021).

60    Cf. *Purgatorio* XXI, 7–13.

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
