# Peer review of "Dante and Siger: An Intellectual Mission Overcoming Error and Authority"

_humanities, doi:10.3390/h13010016_

Round 1
Reviewer 1 Report
Comments and Suggestions for Authors
This is an original and interesting study on Dante's relationship to Siger. This project takes up some neglected issues in the secondary literature and balances them with an interesting and illuminating read of Paradiso.
I found the opening sections of the paper on Dante's relationship with Averroes to need the most attention. These sections seemed more speculative than sourced and the connections between Dante and his alleged Averroism needs to be better demonstrated.
Comments on the Quality of English LanguageThe introduction was a bit challenging to read in regard to the prose. The second half of the essay was much more clearly written.
Author Response
Dear reviewer,
Thank you so much for your precious comments, which I have endeavored to consider thoroughly. Regarding language refinement, I enlisted the services of a professional proofreading and editing team, significantly enhancing the linguistic quality of the article. In addressing the sections concerning the relationship between Dante and Averroism, my aim was to succinctly summarize the existing literature on the subject up to the present day, providing a comprehensive overview of this extensively studied topic. In line with this objective, I alluded to and briefly presented certain theories about Dante's alleged Averroism, avoiding exhaustive detail. My focus was not to argue for an Averroist interpretation of Dante but to delve into specific points of contact with the philosopher Siger, moving beyond the mere identification of a purported Averroistic phase in Dante's work.
Reviewer 2 Report
Comments and Suggestions for Authors
An excellent example of academic writing. The author has tackled a challenging problem and made it accessible to readers. The argument is appropriately nuanced and carefully structured. There is deft engagement with historical context (Aristotle and Siger) as well as with relevant Dante scholarship. I have nothing but praise to offer.
Author Response
Dear reviewer,
Thank you very much for your flattering comment. I'm pleased that my article can appear relevant and, at the same time, readable both within the specialized audience and beyond it.
Reviewer 3 Report
Comments and Suggestions for Authors
The article has some strengths: the ideas are compelling, some parts read well, and it includes a range of sources from noted scholars in the field. However, the writing obscures the author's message in much of the work. Sentences and paragraphs could be restructured, and certain sections made more concise (including the introductory pages). More attention needs to be paid to proper citation: some quotes are not cited at all, and others appear to be translated without explanation. In short, more editing of organization, writing, and citation is needed to strengthen the readability of the article and to allow its ideas to be fully communicated.
Comments on the Quality of English LanguageThe English is often unclear, with many parts reading almost as if they were translated from another language. Some terms give the impression of being used erroneously. These characteristics result in language that is awkward and/or confusing. The article would benefit from more careful editing of the writing.
Author Response
Dear reviewer,
Thank you so much for your precious comments, which I have endeavored to consider thoroughly. Regarding language refinement, I enlisted the services of a professional proofreading and editing team, significantly enhancing the linguistic quality of the article.
Concerning citation, I revised quotes that missed citation and translated into English the ones in Italian.
Reviewer 4 Report
Comments and Suggestions for Authors
Beautifully written and crystal clear, author. Erudite but accessible even to those who are not adepts in medieval philosophy.
Minor edits needed on lines 55, 60, 114 for clarity in wording
Comments on the Quality of English LanguageAgain, very clearly written. Prose is energetic and shows mastery of form and substance.
Author Response
Dear reviewer,
Thank you very much for your flattering comment. I'm pleased that my article can appear relevant and, at the same time, readable both within the specialized audience and beyond it.
I also revised English language through professional proofreading and editing.
Round 2
Reviewer 1 Report
Comments and Suggestions for Authors
I found all of the edits and refinements adequate and I think the article is publishable. Very nice work and I look forward to seeing it in print.
Reviewer 3 Report
Comments and Suggestions for Authors
Second version is a great improvement; the writing is much clearer and the citation issues seem to have been resolved as well.